Microbiology **Spectrum**

# Accuracy of four rapid diagnostic tests (RDTs) for human leptospirosis diagnosis in Indonesia

Farida Handayani,[1,2] Endah Tri Widanarti,[3] Citra W. Kusuma,[4] Ristiyanto Ristiyanto,[5] Amin Soebandrio,[6] Muhammad Hussein Gasem[7,8]

**ABSTRACT** Leptospirosis is an endemic zoonotic disease with protean clinical manifestations caused by pathogenic spirochetes of the *Leptospira* genus. The microscopic agglutination test (MAT) is the gold standard for leptospirosis diagnosis and can only be conducted in a reference laboratory. Therefore, alternative tests, such as the IgM anti-*Leptospira* rapid diagnostic test (RDT), are preferred for general use. In this study, we aimed to compare the accuracy of four products of anti-*Leptospira* IgM detection RDTs, which are available in Indonesia, against the gold standard of *Leptospira* MAT. This study was a diagnostic validation test using bioarchived serum from 364 human serum samples tested by MAT from August to September 2020 in Demak, Central Java, Indonesia. The four products were (i) Fokus *Leptospira*, (ii) Answer *Leptospira*, (iii) SD Bioline *Leptospira* IgG/IgM, and (iv) Uji *Leptospira* IgM, sequentially renamed RDT-1 to RDT-4. Interobserver agreements were analyzed using the kappa value. The diagnostic performance of the four RDTs were compared against MAT results as the gold standard. We also evaluated the combination of two RDTs' performance, which were RDT (1 + 2), RDT (1 + 3), RDT (1 + 4), RDT (2 + 4), and RDT (3 + 4). We found that the kappa coefficients of RDT-1, RDT-2, and RDT-4 were greater than 80%, while RDT-3 had a moderate kappa value of 69.1%. RDT-1, RDT-2, and RDT-4 had moderate to good sensitivities of 78.2%, 74.3%, and 83.6%, respectively, while RDT-3 had the lowest sensitivity at 30.9%. RDT-3 demonstrated the highest specificity. RDT-2 showed the highest predictive value at 75.9%, while RDT-4 showed the highest negative predictive value at 96.9%. In addition, the combination of two RTDs provided better diagnostic performances. The four RDTs performed varied in their ability to diagnose leptospirosis, but only RDT-4 showed a sensitivity of more than 80%. We recommend caution in diagnosing only one RDT result. Testing by other RDTs and confirmation by MAT are strongly recommended.

**IMPORTANCE** The performance of the anti-*Leptospira* IgM antibody rapid diagnostic tests (RDTs) has yet to be evaluated. In this study, we compare the accuracy of these four RDTs available in Indonesia against the gold standard of *Leptospira* microscopic agglutination test (MAT). Utilizing the best performance of RDT in the point of care with limited facilities will be effective since MAT is complicated, laborious, and time-consuming if done at public health centers. The MAT requires maintaining the culture of each live *Leptospira* strain circulating in the region and needs trained personnel.

**KEYWORDS** leptospirosis, rapid diagnostic test, diagnostic accuracy, Indonesia

Leptospirosis is a zoonosis with protean clinical manifestations caused by pathogenic spirochetes of the genus *Leptospira* and has a worldwide distribution. Leptospirosis is thought to be the most widespread zoonosis in the world (1, 2). It is an underreported illness, with no reliable global incidence figures. A systematic review and modeling exercise estimated that more than 1 million human cases occur annually worldwide,

Address correspondence to Amin Soebandrio, asoebandrio@gmail.com.

The authors declare no conflict of interest.

See the funding table on p. 8.

including almost 60,000 deaths yearly (3, 4). It is most prevalent in tropical regions but also occurs in temperate regions. The regions with the highest incidence of infections include South and Southeast Asia, Oceania, the Caribbean, sub-Saharan Africa, and Latin America (3, 4). Humans become infected with contaminated animal urine, tissue, or water. Risk factors include direct animal exposure or activities leading to skin abrasions and water or soil exposure (5, 6).

Leptospirosis is characterized by wide clinical variability, ranging from asymptomatic to severe disease with multiple organ failure that requires hospitalization in an intensive care setting. Common disease manifestations include fever, headache, chills, arthralgia, myalgia, jaundice, diarrhea, nausea, and vomiting (7, 8). Accurate and rapid diagnostic tests for leptospirosis are critical for diagnosing this disease. However, leptospirosis is often misdiagnosed since it is difficult to distinguish from dengue fever, malaria, typhoid, influenza, and other infectious diseases similarly characterized by fever, headache, and myalgia. Therefore, it is necessary to have an easy, convenient, and rapid diagnostic tool to initiate proper and timely management (7).

The gold standard for leptospirosis diagnosis is the microscopic agglutination test (MAT), which detects the presence of potent antibodies against viable *Leptospira*. However, MAT is a complicated test that requires maintaining the culture of each live *Leptospira* strain circulating in the region as testing material. Therefore, enzyme-linked immunosorbent assays (ELISAs), dipstick assays, indirect hemagglutination assays, and rapid diagnostic test (RDT) kits are alternatives for diagnosing leptospirosis. RDTs are helpful solutions for minimizing the need for advanced laboratory tests in these local settings (8, 9). Currently, four commercially rapid diagnostic tests to detect anti-*Leptospira* IgM antibodies are commercially available in Indonesia. The performance of these tests has yet to be elucidated. In this study, we aimed to compare the accuracy of these four RDTs available in Indonesia against the gold standard of *Leptospira* MAT.

## MATERIALS AND METHODS

### Study design

The design of this study is a diagnostic performance test. We compared the RDTs of four commercially available IgM antibodies for leptospirosis patients against the MAT, which is the gold standard. We used archived biological samples collected from the patients. This study was a part of the leptospirosis surveillance implementation activity funded by the PEER Health Program conducted by the Institute for Vector and Reservoir Control Research and Development. Patient enrollment was conducted over 2 months, from August 2020 to September 2020, in Demak District, Central Java, Indonesia. We enrolled patients with acute fever in community health centers (Puskesmas) in Demak District.

The four RDTs tested were (i) Fokus *Leptospira* IgM Cassette, produced by Focus Diagnostic; (ii) Answer *Leptospira* IgG/IgM Combo Rapid Test, produced by CTK Biotech; (iii) SD Bioline *Leptospira* IgG/IgM, produced by Standard Diagnostic; and (iv) Uji *Leptospira* IgM, produced by Pakar Biomedika Indonesia Co. These RDTs were further named RDT-1, RDT-2, RDT-3, and RDT-4, respectively.

### Data collection

We collected data from subjects with acute fever. Demographic data and patient characteristics, including sex, age, occupation, and duration of fever, were recorded. Three hundred sixty-four archived serum specimens that met the inclusion criteria were utilized in this study. The inclusion criterion consisted of having complete demographic and duration of fever data. The samples underwent MATs for leptospirosis and were categorized as MAT positive or negative. After selection, all 364 samples underwent testing with all four RDT brands, as observed by two analysts who assessed the result as positive, negative, or invalid. We considered any positive result reported by any observer as a positive test for the final accuracy analysis.

## Laboratory testing method

The laboratory procedure of the MAT was conducted at the Institute for Vector and Reservoir Control Research and Development, Salatiga, Central Java. In this laboratory, we used 15 leptospiral serovar isolates for the standard protocol for serological diagnosis of leptospirosis. The MAT is a qualitative and quantitative test with high diagnostic specificity and relatively low sensitivity. Sera were screened at a 1:100 dilution, and those showing agglutination were then serially diluted further to determine the titer endpoint. A single serum sample with very high antibody titers ($\geq$1,600) suggested a recent infection. Paired serum titers produce more reliable prognostic information. A fourfold increase in titer or seroconversion to $\geq$1,600 indicates current leptospirosis infection (10).

RDT-1 is a Fokus *Leptospira* IgM Cassette by Focus Diagnostic that is a rapid Leptospirosis-WB that utilizes the principle of immunochromatography, a unique two-site immunoassay on a membrane. Before the test, the sealed pouches in the test kit and the kit components may be stored between 4°C and 30°C for the duration of the shelf-life, as indicated on the pouch. In this RDT, 10 µL of serum/plasma or whole blood was dripped into the sample, and then five drops of the buffer solution were added. The test can be performed within 15 minutes (11).

RDT-2 is an Answer *Leptospira* IgG/IgM Combo Rapid Test by CTK Biotech and is a rapid chromatographic immunological test. Before the test, the sample was heated to room temperature (2°C–80°C). In this RDT, 5 µL of serum, plasma, or whole blood was added to the sample, and then two drops of the buffer solution were added. The test can be performed within 15–20 minutes (12). We only considered the IgM result on this test.

RDT-3 is an SD Bioline *Leptospira* IgG/IgM by Standard Diagnostic and is a lateral flow chromatographic immunoassay used to detect and differentiate IgG or IgM antibodies to *Leptospira interrogans* in human serum, plasma, or whole blood. This RDT used 5 µL–10 µL of serum or plasma or 20 µL of whole blood, and the results were obtained within 15–20 minutes with two or three lines in a cassette defined as positive (13). We only considered the IgM result on this test.

RDT-4 is a Uji *Leptospira* IgM produced by PT Pakar Biomedika Indonesia and is a rapid chromatographic immunological test for detecting human anti-*Leptospira* IgM. Before the test, the sample was heated to room temperature (15°C–30°C). In this rapid test, 5 µL of serum or 5 µL–10 µL of blood was dripped into the sample well, and then 110 µL (three drops) of the buffer solution was added. The test can be performed within 15 minutes. The presence of specific antibodies or positive results will be indicated by the appearance of a purple color on the test line (14).

## Statistical analysis

We present sex, age, occupation, and duration of fever using descriptive statistics to define the baseline characteristics of the subjects. The results are presented in a frequency tabulation. Interobserver variations were described quantitatively with the kappa value. A kappa value >80% was considered to indicate excellent agreement. We determined the test's accuracy by comparing the results of each of the four RDTs against the MAT results as the gold standard. Using the appropriate formula, we calculated the sensitivity, specificity, positive predictive value, negative predictive value, and positive and negative likelihood ratios to present the performance. All the statistical analyses were performed with SPSS version 22 software (IBM SPSS, Chicago, IL).

## RESULTS

### Baseline characteristics

We included 364 archived samples, all were tested for MATs, followed by four leptospirosis RDTs. Most patients included in the sample were male and were aged 18–55 years. The patients' most common occupations were farmer, unskilled laborer, and housewife. A similar proportion of patients was tested if the duration of fever was less than or equal

to 5 days. We found that 55 (15.1%) patients had positive MAT results. Table 1 presents the characteristics of the subjects included in the survey.

## Conformity of the reading results

Two observers read the four leptospirosis RDTs tested. Table 2 presents the reading results of the two observers. The kappa value showed different sensitivity and specificity values between the observers. The kappa coefficients of RDT-1, RDT-2, and RDT-4 showed excellent agreement of more than 80%. The kappa coefficient for RDT-3 was 69.1%. We found that the differences in RDT-1, RDT-2, and RDT-4 scores were minimal, while RDT-3 showed relatively moderate agreement due to the discrepant results.

## Accuracy of RDT against MAT

The accuracy of the RDT compared with that of the MAT as the gold standard is presented in Table 3. RDT-4, RDT-1, and RDT-2 sequentially demonstrated the highest to the lowest sensitivities. RDT-3 has the lowest sensitivity of 30.9%. RDT-3 had the highest specificity, followed by RDT-2, RDT-4, and RDT-1. The RDT with the highest positive predictive value was RDT-2 at 75.9%, followed by RDT-3, RDT-4, and RDT-1. RDT-4 had the highest to lowest negative predictive values, followed by RDT-1, RDT-2, and RDT-3. DT-2, and RDT-3. The highest positive likelihood ratio was demonstrated by RDT-2, which was 17.7, with a good negative likelihood ratio of 0.3. RDT-4 also showed an excellent positive likelihood ratio of 10.3, with a negative likelihood ratio of 0.2. RDT-3 has the highest likelihood ratio negative value at 0.7%.

Figures 1 and 2 show that the sensitivity of RDT-1 compared to RDT-2 and RDT-4 was not significantly different. However, the sensitivity of RDT-3 is significantly (note: reason significantly) lower than that of the other three RDTs. The specificity of RDT-1 was significantly lower than that of RDT-2 and RDT-3. However, RDT-3 is significantly greater than RDT-4. RDT-3 is significantly greater than RDT-1 and RDT-4.

## The combination of two RDT contingency results

We evaluated the performance of two RDT combinations to determine whether this approach is suitable. The result is presented in Table 4. The sensitivity of the RDT

**TABLE 1** Demographic characteristics and fever duration of subjects enrolled for the study

| Characteristics | Total *N* = 364 | % |
|---|---|---|
| Sex, *n* (%) | | |
| Female | 162 | 44.5 |
| Male | 202 | 55.5 |
| Age (years) | | |
| 5–18 | 37 | 10.2 |
| 18–55 | 215 | 59.1 |
| >55 | 112 | 30.7 |
| Occupation, *n* (%) | | |
| Farmers | 123 | 33.8 |
| Unskilled laborers | 64 | 17.6 |
| Housewife | 51 | 14.0 |
| Students | 27 | 7.4 |
| Traders | 22 | 6.0 |
| Others | 77 | 21.2 |
| Duration of fever (*N* 359) | | |
| <5 days | 185 | 51.5 |
| ≥5 days | 174 | 48.5 |
| MAT result | | |
| Positive | 55 | 15.1 |
| Negative | 309 | 84.9 |

**TABLE 2** Interobserver agreement on determining accuracy of the four rapid diagnostic tests

| No. | Brand | Parameter | Observer 1 | Observer 2 | Kappa value |
|-----|-------|-----------|-----------|-----------|-------------|
| 1 | RDT-1 | Sensitivity (%) | 70.9 (58.1–81.8) | 74.5 (62.1–84.8) | 84.9 |
|   |       | Specificity (%) | 84.5 (80.2–88.2) | 85.1 (80.9–88.8) | |
| 2 | RDT-2 | Sensitivity (%) | 56.4 (43.2–69.0) | 67.3 (54.3–78.7) | 88.7 |
|   |       | Specificity (%) | 93.2 (90.0–95.7) | 91.6 (88.1–94.3) | |
| 3 | RDT-3 | Sensitivity (%) | 25.5 (15.2–37.9) | 38.2 (26.1–51.3) | 69.1 |
|   |       | Specificity (%) | 97.1 (94.8–98.6) | 96.1 (93.6–97.9) | |
| 4 | RDT-4 | Sensitivity (%) | 74.5 (62.1–84.8) | 76.4 (64.1–86.2) | 80.0 |
|   |       | Specificity (%)[a] | 89.3 (85.6–92.4) | 83.2 (78.7–87.1) | |

[a]Significant difference between observer 1 and observer 2 with *P*-value <0.05.

combination varied between 19.54% and 93.94%. RDT (3 + 4) showed the highest sensitivity (93.94%). The specificity is quite high, ranging from 80.97% to 97.83%, with RDT (1 + 3) and RDT (2 + 4) having good specificity (97.83% and 90.06%, respectively).

Positive predictive value showed varying results, with the highest value in RDT (1 + 2) at 78.85% and the lowest in RDT (3 + 4) at 25.33%. This shows that although some tests can detect the disease well, the possibility of false-positive results also exists. The negative predictive value was very high for all RDTs, reaching 99.26% for RDT (3 + 4), indicating that if the test result is negative, the patient most likely does not have the disease.

## DISCUSSION

This study evaluated four RDTs for leptospirosis diagnosis that were distributed and commercially available in Indonesia. Agreement between observers varied considerably, with kappa values ranging from 69.1% to 88.7%. The sensitivity of the four tests varied considerably, ranging from 30.9% to 83.6%, but the specificity did not differ much, ranging from 88.4% to 97.7%.

In this study, we enrolled patients at community health centers in Central Java, which mostly has rural settings. Therefore, most of our patients were farmers. We found that the patients were predominantly male in the highly active age range. Although relatively different in the setting, our results are similar to those of another study conducted in a referral hospital in Indonesia. The acute febrile illness study conducted in 2017 revealed that leptospirosis patients aged 18–45 years had a slightly greater male predominance (62.5%) (15). Another previous study also conducted in Yogyakarta, Indonesia, showed a similar result: frequency distribution according to the type of work, with the majority of the population still working as farmers, with the highest percentage, 62.9% (16).

### Interobserver agreement

The interobserver discrepancies in our study should raise awareness of the possible problems in implementation. The three RDTs provided acceptable interobserver

**TABLE 3** Sensitivity, specificity, positive predictive, and negative predictive value of four rapid diagnostic tests

| No. | RDT | MAT | | Sensitivity | Specificity | Pos. predictive value | Neg. predictive value | Pos. likelihood ratio | Neg. likelihood ratio |
|-----|-----|-----|-----|-------------|-------------|----------------------|----------------------|----------------------|----------------------|
| | | Pos *N* = 55 | Neg *N* = 309 | % (95% CI) | % (95% CI) | % (95% CI) | % (95% CI) | % (95% CI) | % (95% CI) |
| 1 | RDT-1 | 43 | 36 | 78.2 | 88.3 | 54.4 | 95.8 | 6.7 | 0.2 |
| | | 12 | 273 | (67.3–89.1) | (84.7–91.9) | (43.4–45.1) | (93.5–98.1) | (3.8–11.9) | (0.2–0.3) |
| 2 | RDT-2 | 41 | 13 | 74.5 | 95.8 | 75.9 | 95.5 | 17.7 | 0.3 |
| | | 14 | 296 | (63.0–86.1) | (93.6–98.0) | (64.5–87.3) | (93.2–97.8) | (10.4–30.2) | (0.2–0.4) |
| 3 | RDT-3 | 17 | 7 | 30.9 | 97.7 | 70.8 | 88.8 | 13.6 | 0.7 |
| | | 38 | 302 | (18.7–43.1) | (96.1–99.4) | (52.6–89.0) | (85.5–92.2) | (8.3–22.4) | (0.4–1.4) |
| 4 | RDT-4 | 46 | 25 | 83.6 | 91.9 | 64.8 | 96.9 | 10.3 | 0.2 |
| | | 9 | 284 | (73.9–93.4) | (88.9–94.9) | (53.7–75.9) | (95.0–98.9) | (5.4–19.9) | (0.1–0.2) |

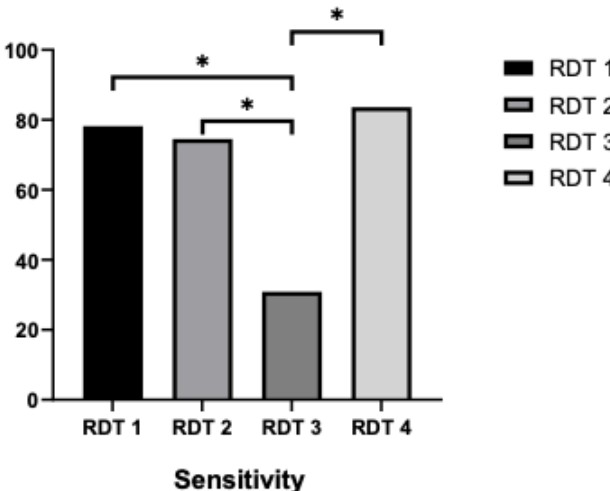

**FIG 1**  Sensitivity of the four leptospirosis IgM rapid diagnostic tests. *P*-value <0.05.

agreement. However, one RDT has a low kappa value. This RDT also has the lowest sensitivity. Previous reports of leptospirosis RDTs have also shown considerable variation in the kappa value, ranging from 56% to 96%, which also occurred in RDTs with the lowest sensitivity (17). Disagreement between readers of RDTs was often caused by a need for more clarity of the T-line result, different times, sample quality (no milking), tool/instrument conditions, and observer ability/skills (18, 19). Our laboratory technician also confirmed that they often find blurred T lines when reading RDT-3 results.

## Accuracy of the tests

The three RDTs had acceptable sensitivities ranging from 74.5% to 84%, except for RDT-3, which showed low sensitivity. All RDTs showed excellent specificity above 88%. The high specificity also correlated with a good positive predictive value above 75.9%, while RDT-4 had the highest negative predictive value. Goris et al., who conducted studies of LeptoTekDriDot, LeptoTek LFA, and Leptocheck-WB, also reported a sensitivity ranging from 75% to 78%, with a high specificity ranging from 95% to 98% (15). In a well-controlled situation, the accuracy of RDTs can be comparable to that of ELISA.

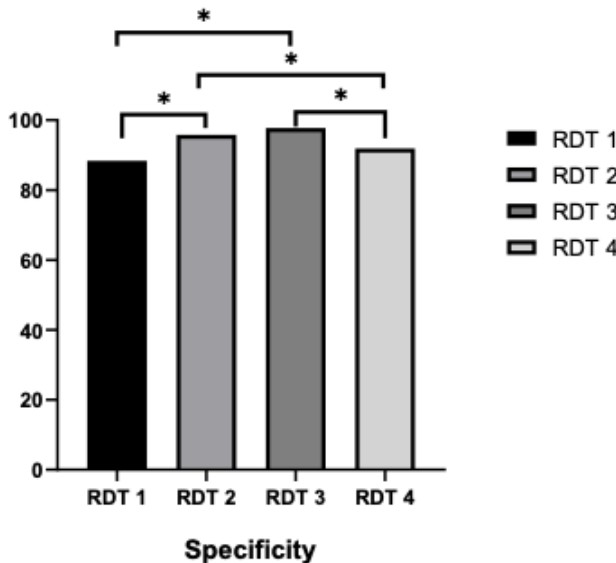

**FIG 2**  Specificity of the four leptospirosis IgM rapid diagnostic tests. *P*-value <0.05.

**TABLE 4** Sensitivity, specificity, positive predictive, and negative predictive value of the two RDT combinations

| | Contingency table analysis 2 × 2 RDT-1–4 | | | | | | | | | |
| | RDT (1 + 2), % | | RDT (1 + 3), % | | RDT (1 + 4), % | | RDT (2 + 4), % | | RDT (3 + 4) | |
| | O1 | O2 | O1 | O2 | O1 | O2 | O1 | O2 | O1 | O2 |
|---|---|---|---|---|---|---|---|---|---|---|
| Single test | | | | | | | | | | |
| Sensitivities | 47.13 | 55.17 | 19.54 | 31.03 | 64.37 | 74.71 | 84.62 | 84.13 | 82.61 | 93.94 |
| Specificities | 96.03 | 94.58 | 97.83 | 97.83 | 93.14 | 89.53 | 90.06 | 86.38 | 83.58 | 80.97 |
| PPV | 78.85 | 76.19 | 73.91 | 81.82 | 74.67 | 69.15 | 58.67 | 56.38 | 25.33 | 32.98 |
| NPV | 85.26 | 87.04 | 79.47 | 81.87 | 89.27 | 91.85 | 97.23 | 96.30 | 98.62 | 99.26 |
| Accuracy | 71.58 | 74.88 | 58.69 | 64.43 | 78.75 | 82.12 | 87.34 | 85.25 | 83.09 | 87.45 |

aO, observer; PPV, positive predictive value; NPV, negative predictive value.

Smits et al. reported the sensitivities of RDT and ELISA, which were 85.8% and 89.3%, respectively, with specificities of 93.6% and 94.2%, respectively (20).

The low sensitivity of RDT-3 has also been reported by Dittrich et al., who also reported that the use of SD Bioline as an RDT showed low sensitivity (21%) and high specificity (94.8%) (17). The very low sensitivity of the SD-IgM assay makes it unsuitable for use as a single diagnostic test unless combined with a secondary test. Combining a high-sensitivity but low-specificity test with a low-sensitivity but high-specificity test may provide a more accurate diagnosis for patients. In addition, we have analyzed the combination of two RDTs. We found that the accuracy of the tests varied but was generally quite good, with the highest value reaching 87.5% for RDT (3 + 4). This suggests that certain combinations of RDTs provide more accurate results in detecting the tested conditions. We also found that in sequential testing, sensitivity decreases significantly compared to single testing, but specificity remains high. In contrast, in simultaneous testing, sensitivity increases dramatically, but specificity decreases. Based on these data, it is important to choose the right combination of RDTs for better diagnosis needs, although more expensive. Simultaneous testing may be a better option to increase detection sensitivity, although it should be noted that there is a potential for increased false-positive results.

## MAT

The MAT is a microscopic method for detecting the agglutination of *Leptospira* bacteria affected by antibodies. The principle of the MAT is that the serum is serially diluted and then mixed with a suspension of live *Leptospira* bacteria at a specific temperature and time. A higher level of dilution, which can still agglutinate bacteria, indicates a high titer of specific antibodies present in the serum. The MAT is the reference test and is often used as the gold standard for evaluating other serological-based *Leptospira* strains because it has high accuracy. In Indonesia, the MAT can be measured at three sites: Bogor, Semarang, and Salatiga. Institute for Vector and Reservoir Control Research and Development (IVRCRD), Salatiga, Central Java, where this study was conducted (https://b2p2vrp.litbang.kemkes.go.id/). This laboratory utilized a panel of 15 serovars that are predicted to be endemic to Indonesia. In this study, 55 (15.1%) patients were positive (10). Our positivity rate was not much different from that of Blacksell's study, which showed that the percentage of patients with an actual leptospirosis infection (as defined by the MAT diagnostic criteria) was 12.4% (21).

## Limitations

The quality of data collection and documentation of information in the community health centers was suboptimal, which may limit the accuracy of fever history collection and patient clinical characterization. A diagnostic test may have different sensitivity and specificity in varying clinical spectra. These limitations underlie our inability to conduct further analysis. It would be interesting to conduct additional studies where we could

combine two tests to increase the yield as well as the specificity of the test. Due to the limited human resources during data collection in the field setting, we could not obtain convalescent sera. Thus, we could not correctly report the true prevalence of leptospirosis because, to confirm this, we need two MATs consisting of acute and convalescent sera. This study was suspended due to the coronavirus disease 2019 pandemic, so we could not obtain more samples than the one reported here.

## Conclusions

RDT is not an ideal diagnostic tool for its accuracy; however, due to its ease of use and minimal laboratory support, it is an essential tool. Most RDTs provide suitable accuracy for detecting *Leptospira* IgM, except for one with low sensitivity. Interobserver agreement can be problematic. However, significant discrepancies occurred in the RDT, which has the lowest accuracy. Thus, its utilization should be further evaluated. Clinicians and laboratory clinicians must find the best options for their setting. The use of more than one RDT, which is the most sensitive and specific test, could be considered. We recommend that a reference laboratory with MAT capability in the country should continuously monitor the quality of these *Leptospira* RDTs.

## ACKNOWLEDGMENTS

Funding was acquired from the WHO, project number/WHO Reference 2020/1033973-0/ Diagnostic Test (ROT) Evaluation for Leptospirosis in Indonesia

The authors whose names are listed above certify that they have no affiliations with or involvement in any organization or entity with any financial interest (such as honoraria; educational grants; participation in speakers' bureaus; membership, employment, consultancies, stock ownership, or other equity interest; or expert testimony or patent-licensing arrangements) or nonfinancial interest (such as personal or professional relationships, affiliations, knowledge, or beliefs) in the subject matter or materials discussed in this manuscript.

## AUTHOR AFFILIATIONS

[1]Doctoral Program in Biomedical Sciences, Faculty of Medicine, University of Indonesia, Jakarta, Indonesia

[2]Eijkman Research Center for Molecular Biology-Indonesian National Research and Innovation Agency (BRIN), Jakarta, Indonesia

[3]Research Center for Care and Control of Infectious Disease (RC3ID) Universitas Padjadjaran, Bandung, Indonesia

[4]Ministry of the Health Republic of Indonesia, NIHRD IVRCRD, Salatiga, Indonesia

[5]Research Center for Public Health and Nutrition, National Research and Innovation Agency (BRIN), Jakarta, Indonesia

[6]Department of Clinical Microbiology, Faculty of Medicine, University of Indonesia-Dr. Cipto Mangunkusumo Hospital, Jakarta, Indonesia

[7]Division of Infectious Diseases and Tropical Medicine, Dr. Kariadi Hospital, Semarang, Indonesia

[8]Faculty of Medicine, Universitas Gunung Jati (UGJ), Cirebon, Indonesia

## AUTHOR ORCIDs

Farida Handayani  http://orcid.org/0000-0001-8540-2700
Amin Soebandrio  http://orcid.org/0000-0002-5856-4808

## FUNDING

| Funder | Grant(s) | Author(s) |
| --- | --- | --- |
| World Health Organization | 2020/1033973-0 | Farida Handayani |

| Funder | Grant(s) | Author(s) |
|--------|----------|-----------|
|  |  | Citra W. Kusuma |
|  |  | Ristiyanto Ristiyanto |

## AUTHOR CONTRIBUTIONS

Farida Handayani, Conceptualization, Formal analysis, Investigation, Validation, Writing – original draft, Writing – review and editing | Endah Tri Widanarti, Formal analysis, Validation, Writing – original draft | Citra W. Kusuma, Data curation, Investigation | Ristiyanto Ristiyanto, Conceptualization, Formal analysis, Methodology, Supervision, Validation, Writing – review and editing | Amin Soebandrio, Conceptualization, Methodology, Supervision, Validation, Writing – review and editing | Muhammad Hussein Gasem, Conceptualization, Methodology, Supervision, Validation, Writing – review and editing

## ETHICS APPROVAL

This research received approval from the National Institute of Health Research and Development Ethics Committee, Ministry of Health, Indonesia No. LB.02.01/2/KE.090/2019.

## ADDITIONAL FILES

The following material is available online.

Open Peer Review

**PEER REVIEW HISTORY (review-history.pdf).** An accounting of the reviewer comments and feedback.

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
