## [Reviewer comments · Microbiology Spectrum]

Microbiology Spectrum

Accuracy of Four Rapid Diagnostic Tests (RDTs) for Human Leptospirosis Diagnosis in Indonesia

Farida Handayani, Endah Widanarti, Citra Kusuma, RISTIYANTO RISTIYANTO, Amin Soebandrio, and Muhammad GASEM

Corresponding Author(s): Amin Soebandrio, Universitas Indonesia

Review Timeline:

Submission Date:	July 2, 2024
Editorial Decision:	September 1, 2024
Revision Received:	January 3, 2025
Accepted:	January 30, 2025

Editor: Remi Charrel

Reviewer(s): The reviewers have opted to remain anonymous.

Transaction Report:

DOI: <https://doi.org/10.1128/spectrum.01524-24>

Re: Spectrum01524-24 (Accuracy of Four Rapid Diagnostic Tests (RDTs) for Human Leptospirosis Diagnosis in Indonesia)

Dear Prof. Amin Soebandrio:

Thank you for the privilege of reviewing your work. Below you will find my comments, instructions from the Spectrum editorial office, and the reviewer comments.

From the performances of the 4 RTD, it is difficult to recommend a single one between RTD1, 2 and 4: basically, all negative results should be confirmed by MAT ref test. in contrast, the combination of two RTD might provide better performances: so it would be interesting to complete the analysis by analysing combination of 2 RTD in order to define whether this approach, although more expensive, is suitable. a table including all the results (RTD1-4 and MAT) for the 364 samples is needed in order to support the combinatory analysis aforementioned

Revision Guidelines

Sincerely,
Remi Charrel
Editor
Microbiology Spectrum

Re: Spectrum01524-24R1 (Accuracy of Four Rapid Diagnostic Tests (RDTs) for Human Leptospirosis Diagnosis in Indonesia)

Dear Prof. Amin Soebandrio:

Your manuscript has been accepted, and I am forwarding it to the ASM production staff for publication. Your paper will first be checked to make sure all elements meet the technical requirements. ASM staff will contact you if anything needs to be revised before copyediting and production can begin. Otherwise, you will be notified when your proofs are ready to be viewed.

Sincerely,
Remi Charrel
Editor
Microbiology Spectrum